# Identification and Elimination of the Clinically Relevant Multi-Resistant Environmental Bacteria *Ralstonia insidiosa* in Primary Cell Culture

**DOI:** 10.3390/microorganisms8101599

**Published:** 2020-10-17

**Authors:** Dennis Nurjadi, Sébastien Boutin, Katja Schmidt, Melinda Ahmels, Daniel Hasche

**Affiliations:** 1Department of Infectious Diseases, Medical Microbiology and Hygiene, Heidelberg University Hospital, Im Neuenheimer Feld 324, 69120 Heidelberg, Germany; Dennis.Nurjadi@med.uni-heidelberg.de (D.N.); Sebastien.Boutin@med.uni-heidelberg.de (S.B.); 2German Cancer Research Center (DKFZ), Division of Microbiological Diagnostics (W440), Im Neuenheimer Feld 280, 69120 Heidelberg, Germany; katja.schmidt@dkfz-heidelberg.de; 3German Cancer Research Center (DKFZ), Division of Viral Transformation Mechanisms (F030), Im Neuenheimer Feld 242, 69120 Heidelberg, Germany; m.ahmels@dkfz-heidelberg.de

**Keywords:** bacterial contamination, primary cell culture, *Ralstonia insidiosa*, multidrug resistance

## Abstract

In times of spreading multidrug-resistant bacteria, species identification and decontamination of cell cultures can be challenging. Here, we describe a mobile cell culture contaminant with “black dot”-like microscopic appearance in newly established irreplaceable hybridoma cell lines and its identification. Using 16S rRNA gene sequencing, species-specific PCRs, whole genome sequencing (WGS), and MALDI-TOF mass spectrometry, the contaminant was identified as the ubiquitous environmental and clinically relevant Gram-negative bacterium *Ralstonia insidiosa* (*R. insidiosa*), a strong biofilm producer. Further characterizations by transmission electron microscopy (TEM) and biochemical API test were not conclusive. Whole genome sequencing of our *R. insidiosa* isolate revealed numerous drug-resistance determinants. Genome-wide comparison to other *Ralstonia species* could not unambiguously designate our isolate to *R. insidiosa* (<95% average nucleotide identity) suggesting a potential novel species or subspecies, closely related to *R. insidiosa* and *R. pickettii*. After determining the antibiotic susceptibility profile, the hybridoma cell culture was successfully decontaminated with ciprofloxacin without affecting antibody production.

## 1. Introduction

Contamination with bacteria, yeast, fungi, or other cell lines is a ubiquitous danger for cell cultures, despite aseptic work conditions, autoclaved equipment, and sterile media. While many bacterial contaminations are clearly visible, for instance by massive turbidity and color change of the acidified medium, infiltration and cross-contamination by another cell line often remains undetected [1,2] until subjection to authentication, e.g., by short tandem repeat (STR) profiling [3], as required by more and more journals. Contaminations usually result in the loss of the cell line or misinterpretation of experimental data, for instance due to stealing of nutrients and unwanted activation of signaling cascades [4,5]. Intracellular mycoplasma, which affect up to 30% of cell lines [6] or slow-growing extracellular bacteria that, contrary to the wide-spread opinion, do not cause medium color change and therefore remain unnoticed pose a particular danger. Published case reports are limited, especially regarding the identification of unknown bacterial contaminants, described as “swimming black dots” [7,8]. However, many online science forums contain questions and discussions about unidentified contaminations that appear as extracellular black dots propagated via normal cell splitting that may even tend to move [9,10,11,12,13]. Many replying experimenters are still of the opinion that (a) bacterial contamination would cause a color change, and that it is therefore debris driven by Brownian motion, or (b) the cells should simply be discarded. Some good sounding ideas like extensive washing, filtration through strainers, or centrifugation are impracticable, since they will not remove all bacteria. Although several antibiotics are available to clean the cells, many bacteria are multidrug-resistant (MDR), especially to the commonly used combination penicillin/streptomycin. Therefore, the use of uncommon antibiotics that have to be determined a priori by sensitivity testing may be indicated. In most cases, especially when cryopreserved backups or commercial vendors are available, such an effort may be beyond the benefits. Nevertheless, in some cases that involve irreplaceable cells derived from primary sources [14] or precious hybridoma clones that were generated in a laborious and time consuming procedure and produce unique antibodies [15], a cleanup of the cells may be worthwhile.

Irrespective of the laboratory setting, antibiotic resistances are spreading worldwide, also in hospitals. Nosocomial infections represent an emerging problem especially in intensive care units where seriously ill, highly vulnerable and often immunocompromised patients are treated. Such infections have been clearly linked to the use of devices as endotracheal intubation, urinary catheters, and central venous catheters [16]. With the percentage of infections with MDR Gram-negative bacteria increasing, treatment options are limited [17]. Strong biofilm producers like *Acinetobacter baumannii* [18] or *Ralstonia* species are hard to fight, as they are well-protected against antimicrobials. The bacterial genus *Ralstonia* includes ubiquitous environmental bacteria known as phytopathogens (*R. solanacearum*) [19] and are a growing threat in hospitals, e.g., for patients under mechanical ventilation or suffering from cystic fibrosis (*R. pickettii*, *R. insidiosa*, and *R. mannitolilytica*) [20,21,22]. They can survive under low-nutrient conditions even in laboratory and hospital water supplies. Further, in particular, *R. insidiosa* as a strong biofilm producer facilitates the settlement of other bacteria [23,24].

Here we report the contamination of unique hybridoma clones with an unknown bacterial contaminant and its identification as a MDR strain of the clinically relevant *R. insidiosa*.

## 2. Materials and Methods

### 2.1. Hybridoma Culture Conditions and Amplification of Bacterial Contaminant

Hybridoma cells obtained from a fusion of murine Sp2/0-Ag 14 cells and *Mastomys coucha* splenocytes [25] were cultured in 6-well plates in RPMI-1640 (Sigma-Aldrich, catalogue no 8758, Munich, Germany) supplemented with 10% (*v*/*v*) heat-inactivated fetal calf serum (FCS; Gibco, Thermo Fisher Scientific, Waltham, MA, USA), 100 units/mL penicillin and 100 mg/mL streptomycin in at 37 °C and 5% CO_2_.

### 2.2. Extraction of Bacteria from the Cell Culture and Pure Culture

Cell culture medium of a contaminated well was transferred without hybridoma cells to a new well and cultured until the medium was turbid. The medium was resuspended to detach bacteria from the bottom and 100 µL were centrifuged for 5 min at 500× *g* and resuspended in 25 µL PCR-grade water for PCRs. Additional aliquots were centrifuged, resuspended in Dulbecco’s Phosphate Buffered Saline (DPBS; Sigma-Aldrich, catalogue no R8537, Munich, Germany) and subjected to TEM or to bacterial culture on Columbia blood agar plates with sheep blood (Oxoid, Thermo Fisher Diagnostics GmbH Microbiology, Wesel, Germany). Agar plates were incubated at 37 °C and 5% CO_2_ until bacteria were sub-cultured 48 h later.

### 2.3. Amplification and Sequencing of the Bacterial 16S RNA Gene

Bacterial 16S rRNA gene was partially amplified with the universal bacterial 16S rRNA forward primer 27F (5′-AGAGTTTGATCCTGGCTCAG-3′) and the reverse primers 1492R (5′-GGTTACCTTGTTACGACTT-3′) or 16s907 (5′-CCGTCAATTCMTTTRAGTTT-3′) [26,27] using 2× DreamTaq^®^ Polymerase Master Mix (Thermo Fisher Scientific, Waltham, MA, USA):10 µL 2× Master Mix, 1 µL Primer Mix (20 µM), 7 µL H_2_O, 2 µL sample. Thermal cycling conditions for PCRs were a primary denaturation step at 95 °C for 3 min, followed by 30 cycles of 30 s at 95 °C, 30 s at 55 °C, 45 s at 72 °C, and a final extension step of 10 min at 72 °C. DNA fragments were separated by electrophoresis on a 1.2% agarose gel, stained and visualized by UV light. PCR products were extracted from agarose gel with QIAquick^®^ Gel Extraction Kit (Qiagen GmbH, Hilden, Germany) and sequenced with 27F primer by Eurofins Genomics Germany GmbH (Ebersberg, Germany).

### 2.4. Specific Identification of R. insidiosa by PCR

Specific identification of *R. insidiosa* was based on PCR with three primer pairs: 1. Rp-F1 (5′-ATGATCTAGCTTGCTAGATTGAT-3′) & Rp-R1 (5′-ACTGATCGTCGCCTTGGTG-3′) for the detection of *R. pickettii*, 2. Rm-F1 (5′-GGGAAAGCTTGCTTTCCTGCC-3′) & Rm-R1 (5′-TCCGGGTATTAACCAGAGCCAT-3′) for the detection of *R. mannitolilytica* and 3. Rp-F1 & R38R1 (5′-CACACCTAATATTAGTAAGTGCG-3′) for the detection of *R. insidiosa* [22,28] using 2× DreamTaq^®^ Polymerase Master Mix (Thermo Fisher Scientific, Waltham, MA, USA) and 2 µL sample and the thermal scheme described above. DNA fragments were separated by agarose gel electrophoresis, stained and visualized by UV light. *R. insidiosa* is detected with primer combination Rp-F1/R38R1, but also with Rp-F1/Rp-R1, while it is not detectable with Rm-F1/Rm-R1 [28].

### 2.5. API Test

API^®^ 20 NE test (BioMérieux Deutschland GmbH, Nürtingen, Germany) was performed according to the manufacturers’ protocol based on current microbiology standards. After incubation at 30 °C for 24 h and 48 h, the biochemical reactions and bacterial growth were analyzed according to the manual to determine the numerical profile, which allowed identification using the apiweb^TM^ identification software.

### 2.6. Gram Stain

Gram staining was performed as previously described [29].

### 2.7. Light Microscopy

Pictures of contaminated cell cultures were taken with an EVOS XL Core Imaging System and 20× or 40× long working distance (LWD) objectives (Life Technologies Corporation, Bothell, WA, USA).

### 2.8. Transmission Electron Microscopy (TEM)

Bacteria were fixed with buffered aldehyde solution (4% formaldehyde, 2% glutaraldehyde, 1 mM MgCl_2_, 1 mM CaCl_2_ in 100 mM calcium cacodylate, pH7.1), followed by post-fixation in buffered 1% OsO_4_, graded dehydration with ethanol and resin-embedding in epoxide (12 g glycid ether, 6.5 g N,N-dimethylacetamide (NMA), 6.5 g dodecenylsuccinic anhydride (DDSA), 400 mL 2,4,6-Tris(dimethylaminomethyl)phenol (DMP30), all from Serva, Heidelberg, Germany). Ultrathin sections at nominal thickness 60 nm and contrast-stained with lead-citrate and uranyl acetate were observed in a Zeiss EM 910 at 100 kV (Carl Zeiss, Oberkochen, Germany).

### 2.9. MALDI-TOF MS and Antibiotic Susceptibility Testing

For analysis of the species, bacterial pure culture was subjected to MALDI-TOF (matrix-assisted laser desorption-ionization time of flight) mass spectrometry (MS) (Bruker Diagnostics, Germany) as a duplicate and processed by the routine diagnostic laboratory according to current microbiological diagnostic standards. Species identification was performed as duplicates and a score of >2.0 was considered reliable to the species level [30]. Antibiotic susceptibility testing was performed using the VITEK^®^2 (Biomérieux, Nürtingen, Germany) with the AST-N389 panel and interpreted based on the EUCAST clinical breakpoints (v10.0) for *Pseudomonas* spp. There are no breakpoints for bacteria of the genus *Ralstonia*. Since *Ralstonia* spp. were once assigned to the genus *Pseudomonas* [31], clinical breakpoints for *Pseudomonas* spp. would be the most suitable. MIC (minimal inhibitory concentration) for imipenem and meropenem were confirmed by Etest (Liofilchem, Roseto degli Abruzzi (TE), Italy).

### 2.10. Carbapenem Inactivation Assay

The production of carbapenem hydrolyzing enzyme was detected using a meropenem disk in a carbapenem inactivation assay, as published elsewhere [32]. Briefly, a loopful (1 µL) of an overnight culture of *Escherichia coli* ATCC^®^ 25922 (negative control), Verona-integron-metallo-betalactamase-1 (VIM-1) producing *Citrobacter amalonaticus* KE3510 (positive control) [33] and our *R. insidiosa* isolate were each suspended in 2 mL tryptic soy broth (TSB). A meropenem disk (10 µg, Sensidisc, BD Diagnostics, Heidelberg, Germany) was added into each bacterial suspension after short vortexing for 10–15 s and incubated for 4 h at 37 °C in ambient air. After this incubation period, each disk was removed from the TSB, excess liquid was removed and placed onto a Mueller-Hinton plate, which was inoculated with the multi-susceptible *E. coli* ATCC^®^ 25922 (0.5 McFarland standard). Reduction in zone of inhibition after 18–24 h incubation at 37 °C compared to the meropenem disk incubated with the negative control is an indication of meropenem inactivation.

### 2.11. Whole Genome Sequencing and Data Analysis

Genomic DNA was extracted from overnight bacterial culture using the DNeasy Blood and Tissue Minikit (Qiagen GmbH, Hilden, Germany). Standard genomic library was prepared from the bacterial DNA and sequenced with the Illumina MiSeq platform (2 × 300 bp paired end), as described elsewhere [34]. For quality control, raw sequences were trimmed using Sickle 1.33 (parameters, q  >  30; 1  >  45). Clean reads were assembled with spades 3.13 with the option –careful and –only-assembler [35]. Obtained contigs were curated for length (>1000 bp) and coverage (>10×). Sequence was annotated using Prokka 1.14.1 (based on Genetic Code Table 11). Resistance genes were found using Abricate 0.8.13. Briefly, the draft genome was mapped to the database of CARD, NCBI AMRFinderPlus, Resfinder and ARG-ANNOT [36,37,38,39] and hits with a minimum identity of 90% and a minimum coverage of 80% were considered as AMR genes present in our draft genome. The sequence was uploaded to the NCBI database under Bioproject Accession PRJNA661395.

### 2.12. Phylogenetic Tree

The phylogenetic tree was constructed using the Gubbins algorithm [40] with publicly available complete genomes of *Ralstonia* spp. from the Refseq database. Briefly, the core genome was calculated using Roary, and contains every gene present in all the species (378 genes). Next, recombination events are detected iteratively using the default setting (--min_snps 3, --converge_method weighted_robinson_foulds, --iterations 5). The tree was constructed using RAxML. Using the same procedure, a second tree was built using complete and draft genomes of *R. insidiosa*. The core genome was bigger due to close relatedness (3497 genes).

## 3. Results

### 3.1. Detection of a Cell Culture Contamination

Hybridoma clones were generated by fusion of splenocytes from the rodent *Mastomys coucha* [25] and the murine Sp2/0-Ag 14 cell line, repeatedly screened for the production of specific antibodies and subcloned via limited dilution to ensure monoclonality [15]. Each subcloning required addition of freshly and aseptically isolated feeder splenocytes. After the third and final subcloning, in 11 of the resulting 36 hybridoma lines, tiny black dots that seemed to be mobile could be observed under the microscope (Figure 1A). Although mycoplasma would not be visible, a PCR-based test was performed with a negative result. After larger attached dots without motility appeared and the bottom of the plate was covered by a biofilm also visible under the light microscope (Figure 1B). These black dots multiplied and were carried along with usual splitting of the hybridoma cells. To exclude that it was cellular debris, the supernatant of the culture was transferred to a new 6-well plate and cultured under conditions as before. Despite a very low amount of remaining hybridoma cells, the number of these mobile black dots increased. Further, passaging was possible until no hybridoma cells but only the black dots were left, indicating a living contaminant. Indeed, visualization via transmission electron microscope (TEM) showed that these dots were prokaryotes (Figure 1C,D).

### 3.2. Identification of Bacterial Species

Although the hybridoma cells were cultured in medium containing penicillin/streptomycin, the contaminant seemed to be a bacterium. Therefore, we ran several approaches to identify the species. We subjected an aliquot of the contaminated culture to PCR for 16S rRNA gene amplification via two primer combinations suitable for identification of a broad range of bacteria as previously reported (Figure 2A) [27]. Sequencing of the products revealed 99.6% (primers 27F & 1492R) and 98.48% (primers 27F & 16s907) sequence homology to *R. insidiosa*. The genus *Ralstonia* comprises several ubiquitous environmental bacterial species. While some are known as phytopathogens, some members are also associated with hospital infections [41]. Due to their clinical importance, primers have been previously established for the distinction between *R. pickettii*, *R. insidiosa*, and *R. mannitolilytica* [22,28]. While amplification with the primer combination Rm-F1/Rm-R1 was negative, primers Rp-F1/Rp-R1 and F1/R38R1 resulted in specific products, which confirmed that our bacterial contaminant was *R. insidiosa* (Figure 2B), a Gram-negative rod-like bacterium (Figure 2C). The bacterium was cultured on Columbia blood agar and biochemically tested with the API^®^ 20 NE test system (Figure 2D), which identifies bacteria based on their biochemical properties determined in 20 tests (the analytical profile index) [42]. The resulting analytical profile index of 0050555 was compared to the APIWEB^TM^ database, which indicated *R. pickettii* with a confidence of 56.3% as it did not contain *R. insidiosa*. Due to this discrepancy in the identification, an additional MALDI-TOF MS analysis was performed and again clearly identified *R. insidiosa* with scores of 2.22 and 2.18 (a score of >2.0 is considered a reliable identification to the species level [30]).

### 3.3. Antibiotic Susceptibility Profile of R. insidiosa

Since the contaminant could not be eradicated with antibiotics commonly used in a biomolecular laboratory (penicillin/streptomycin, ampicillin, gentamycin), we determined the antibiotic susceptibility profile of the *R. insidiosa* contamination to finally cleanup the hybridoma cells. Interestingly, this strain of *R. insidiosa* was indeed resistant to multiple antibiotics of different classes (Table 1), including carbapenems and gentamicin. The genotypic resistance determinants are summarized in Appendix A. Therefore, based on the susceptibility profile, a decontamination attempt with 0.5 mg/mL ciprofloxacin was performed, which successfully eliminated the contamination without affecting hybridoma cell growth or antibody production.

### 3.4. Genomic Characteristics of R. insidiosa

Despite their clinical relevance as opportunistic multidrug-resistant global pathogens [20] and their emergence as causative agents of nosocomial infections [41,43,44,45], publicly available genome sequences of *R. insidiosa* are rare. Therefore, we sequenced our isolate to perform a phylogenetical comparison to published *Ralstonia* spp. sequences to investigate the underlying mechanism for the multidrug-resistance. The phylogenetic tree (Figure 2E) shows that our strain clusters closely with *R. insidiosa*, validating the results obtained by PCR and MALDI-TOF MS. Furthermore, a second phylogenetic tree focusing on *R. insidiosa* showed that our strain clusters in a branch distant from the representative genomes (ATCC^®^ 49129) and is closely related to strains isolated in the International Space Station (ISS) (NCBI Bioproject Accession: PRJNA493516). Despite an MDR phenotype, analysis of the whole genome sequencing (WGS) data did not detect an abundance of antimicrobial resistance (AMR) genes. Only two class D beta-lactamase encoding genes, bla_OXA-573_ and bla_OXA-574_, were identified. We did not find other resistance genes for aminoglycosides, which would explain the gentamicin resistance. However, several genes with around 70% identity to efflux pump systems found in non-fermenting Gram-negative bacteria were identified and may be responsible for the phenotypic resistance to gentamicin (Appendix A). However, our *R. insidiosa* isolate could not hydrolyze meropenem, as indicated by the negative result of the carbapenem inactivation assay (Appendix A), hence the presence of these genes alone could not explain the carbapenem resistance [41]. This observation is consistent with the result of cloning experiments by Fang et al., which did not demonstrate cabapenemase activity for bla_OXA-573_ and bla_OXA-574_ [41]. In addition, OXA-573 belongs to the OXA-60 family, which is intrinsically present in *R. pickettii* and only exhibits hydrolyzing activity for imipenem but not for meropenem [46]. Nevertheless, the annotation of the genome showed that an efflux RND transporter system and multiple multidrug export/resistance genes are present in this strain, which may explain the resistance to carbapenems. However, without a susceptible strain to compare, we were not able to pinpoint the mutation or gene involved in the carbapenem resistance.

## 4. Discussion

Bacterial contaminations represent a ubiquitous danger for cell cultures, especially when multidrug-resistant bacteria are involved. While contaminated cultures are usually discarded, this is not possible for precious cell lines. Here, we presented several lab techniques to identify the contaminant of our unique hybridoma clones and its antibiotic susceptibility profile in order to enable a decontamination of the cell lines. The contaminant was identified as rod-shaped, Gram-negative, non-fermenting bacterium *R. insidiosa* via PCR, 16S rRNA sequencing and MALDI-TOF MS. As experienced before [44], the API^®^ test was not conclusive. The identification of bacteria of the genus *Ralstonia* on the species level can be challenging, and misclassifications are possible. Due to the low discriminatory power and possibly incomplete database entries, biochemical identification tests are not reliable for correct species identification, as demonstrated in our case by the inconclusive API^®^ test. Consistent with the literature, both PCR-based methods and MALDI-TOF MS could correctly identify our *R. insidiosa* isolate to the species level, while the latter is probably the fastest and most economical identification method [28,47]. Furthermore, whole genome sequencing (WGS) confirms that our strain is related to *R. insidiosa*. However, it clustered outside of the representative branch, which may indicate that our strain belongs to a species phylogenetically between *R. insidiosa* and *R. pickettii*.

Generally, *Ralstonia* spp. are considered as robust and tenacious organisms, which thrive well in moist environments [21,48]. Bacteria of the genus *Ralstonia* are commonly associated with contaminations of medical devices, water and even blood culture bottles [20,49,50,51,52]. In addition, colonization and infection with these bacteria are encountered in the immunocompromised and patients on mechanical ventilators [21,28,44]. Along with *R. pickettii*, *R. mannitolilytica*, and *R. insidiosa* are the most relevant pathogens of the *Ralstonia* genus and are considered as emerging opportunistic pathogens with increasing clinical relevance [41]. Infections with *R. insidiosa* can range from minor superficial infections to more severe systemic infections, i.e., bacteremia, even in newborns [28,41,44,45].

Initial decontamination attempts with penicillin/streptomycin, the most commonly used antibiotic combination in cell culture experiments, and even gentamicin, failed to successfully eliminate the contaminant. Indeed, trimethoprim-sulfamethoxazole and quinolones have been suggested as one of the best options to treat *Ralstonia*. infections as suggested by in vitro susceptibility testing of diverse species [51], which have been described as strong biofilm producers [24,53]. The presence of a biofilm on plastic surfaces may provide a protective environment for contaminants and reduces the efficacy of certain antibiotics [54]. In our setup, we managed to successfully eliminate the contaminant by using the quinolone ciprofloxacin, which can penetrate biofilms [55], and ultimately decontaminated the hybridoma cells without affecting their antibody production.

Systematic antibiotic susceptibility data for *Ralstonia* spp. are scarce. Nevertheless, several studies and case reports suggested that most *Ralstonia* spp. are susceptible to commonly used antibiotics [51,56,57]. Our *R. insidiosa* isolate exhibited high-level resistance to all beta-lactams including carbapenems (Table 1). The emergence of MDR *R. insidiosa* in clinical isolates was recently reported from a Chinese tertiary hospital and the carbapenem resistance was attributed to the presence of a class D OXA-type beta-lactamase (bla_OXA-570_) [41]. In our isolate, we did not find any evidence for the presence of carbapenem-hydrolyzing enzymes, as shown by a negative in vitro carbapenem inactivation assay. WGS identified the presence of antimicrobial resistance (AMR) genes bla_OXA-573_ and bla_OXA-574_ in our *R. insidiosa* isolate and in the ATCC^®^ 49129 *R. insidiosa* type strain, as well as in 9 of the 12 strains listed in the Refseq database. Both AMR genes are similar to the chromosomally encoded inducible imipenem-hydrolyzing class D beta-lactamases, bla_OXA-22_ and bla_OXA-60_, which are widespread in *R. pickettii* [46], but these AMR genes cannot explain the meropenem resistance. Due to the lack of large epidemiological and molecular data, it is not known whether bla_OXA-573_ and bla_OXA-574_ are intrinsic to *R. insidiosa*. In our isolate, the reduced susceptibility to meropenem may be mediated by other mechanisms, such as drug efflux pumps or porin loss [58]. Genetically, our R. insidiosa strain clusters with other sequenced and publicly available *R. insidiosa* isolates. Interestingly, our *R. insidiosa* is closely related (single nucleotide polymorphism <20 nt over the core genome containing 253628 polymorphic sites) to the unpublished *R. insidiosa* isolates from the International Space Station (ISS) (NCBI Bioproject Accession: PRJNA493516). The average nucleotide identity between our strain and the strains S42, S44, S58, and S59 were 99.97%, 99.35%, 99.33%, and 99.35%, respectively. However, there is no background information on the ISS *R. insidiosa* isolates, so that it is not possible to draw any conclusions from this observation.

Even though our MDR *R. insidiosa* strain was not isolated from a clinical sample, cumulating reports on the emergence of multidrug resistance in *Ralstonia* spp. in the context of human infections raise some concern. Due to the lack of systematic epidemiological and resistance data, it is neither possible to estimate the prevalence of AMR genes, nor assess the clinical relevance of their presence. Nonetheless, it appears that AMR genes in *Ralstonia* spp. may be more widespread than previously anticipated and warrant further investigations.

## Figures and Tables

**Figure 1 microorganisms-08-01599-f001:**
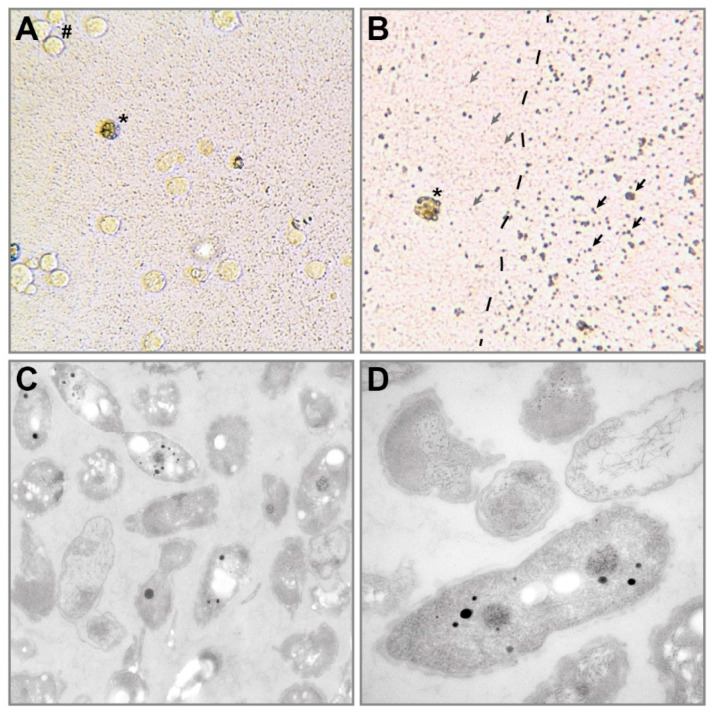
Microscopic observation of the contamination. (**A**) Contaminated hybridoma culture visualized with a 20× objective. (**B**) Propagated contamination visualized with a 40× objective. Only a few dead hybridoma cells were left, while bacteria produced a biofilm. #: vital hybridoma cells, *: dead hybridoma cell, grey arrows: swimming bacteria, black arrows: attached bacteria in a biofilm. (**C**) TEM picture of the bacteria at 20,000× magnification. (**D**) TEM picture of the bacteria at 40,000× magnification.

**Figure 2 microorganisms-08-01599-f002:**
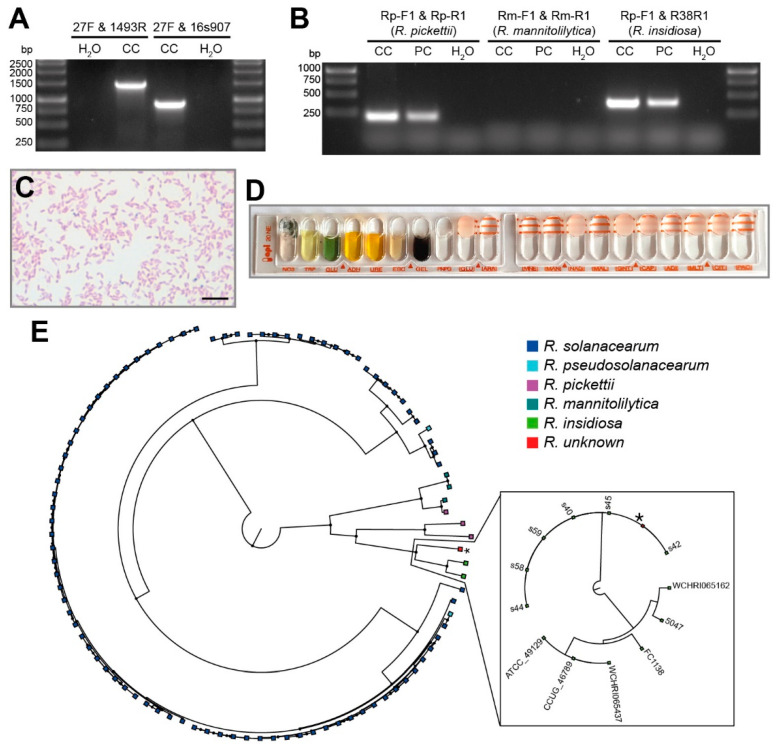
PCR, biochemical and phylogenetic analyses of the contamination. (**A**) 16S rRNA gene amplification for sequencing with two different primer combinations (CC: sample from contaminated cell culture; H_2_O: water control). (**B**) Specific identification of *Ralstonia* species via PCR (CC: sample from contaminated cell culture; PC: pure culture of *R. insidiosa*; H_2_O: water control). (**C**) Gram stain of *R. insidiosa* pure culture originating from the cell culture visualized with a 100× objective (scale bar: 5 µm). (**D**) Image of the API^®^ 20 NE test strip 48 h after inoculation. (**E**) Phylogenetic tree based on the core genome (378 genes) of all complete genomes of *Ralstonia* spp. found in the Refseq database. The framed tree is based on the core genome (3497 genes) of complete and draft genomes of *R. insidiosa* strains.

**Table 1 microorganisms-08-01599-t001:** Antibiotic susceptibility profile of the *R. insidiosa* isolate.

Antibiotic Class	Antibiotic	MIC (Minimal Inhibitory Concentration [µg/mL]) ^a^
Ureidopenicillin/beta lactamase inhibitor	Piperacillin/Tazobactam	≥128 (R)
Cephalosporin	Ceftazidim	32 (R)
Cephalosporin	Cefepim	16 (R)
Carbapenem	Imipenem	>32 (R) ^b^
Carbapenem	Meropenem	8 (I) ^c^
Quinolone	Ciprofloxacin	0.5 (S)
Quinolone	Levofloxacin	1 (S)
Aminoglycoside	Gentamycin	≥16 (R)

(R): Resistant; (I): Intermediate; (S): Sensitive; ^a^ MIC was tested using VITEK^®^2, interpretation based on EUCAST clinical breakpoints v10.0 for *Pseudomonas* spp. (*Ralstonia* spp. were previously classified as *Pseudomonas* spp. *prior* to reclassification to the genus *Ralstonia*. Breakpoints are not available for *Ralstonia* spp.); ^b^ Antibiotic susceptibility testing was performed by Etest; ^c^ MIC confirmed by E-test (8 µg/mL).

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
