# Peer review of "Identification and Elimination of the Clinically Relevant Multi-Resistant Environmental Bacteria Ralstonia insidiosa in Primary Cell Culture"

_microorganisms, 2020, doi:10.3390/microorganisms8101599_

Round 1

Reviewer 1 Report

Nurjadi et al. present an interesting work that resulted in the identification of a living contaminant that they observed in their irreplaceable cell lines. Through PCR, 16S rRNA sequencing and mass spectrometry, they identify the bacterial species in question. Importantly, after identifying the unknown organism, they successfully eradicate it following antibiotic susceptibility testing.

The manuscript is well written, and is clearly explained. This work will be of practical use to those whose research involves cultivating rare or unique cell lines. The methods section could use some more details, the “not shown” data should be included, and there are a few grammatical mistakes. I have the following comments and suggestions.

Line 34: Instead of cell lines, “microorganisms” is a more appropriate word here.

Line 37: Here, “another cell line” gets confusing with cell line being mentioned again later in that sentence.  Maybe use “others,” implying other bacterial species.

Line 58: Change “even” to “especially.”

Line 59: Remove “especially.”

Line 62: Change “While” to “With.”

Line 66: Change to: “…and are a growing threat…”

Line 68-69: It would be clearer to split this long sentence into two. Place a period after “water supplies.”  The next sentence would read: “R. insidiosa is a strong biofilm producer, which facilitates the….”

Line 71-72: Change to “an MDR strain of the clinically…” and remove “with a MDR susceptibility profile.”

Line 77: Define FCS.

Line 80: Remove “6-.”

Line 82: Change “Further” to “Additional.” Define PBS and composition used.

Line 91: Remove “based on.”

Line 109: Change “according to” to “based on current…”

Line 115: Add a few more details to this section. How were samples obtained (growth conditions, etc.), and how were slides prepared?

Line 122: What is NMA, DDSA and DMP30?

Line 126-130: Reference?

Line 135-138: This section is too vague, so please briefly describe this assay.

Section 2.11: Please provide more details on the data manipulation and analysis, including statistical analysis performed and number of replicates.

Section 2.12: This section is also vague. Please add more details to how the trees were constructed.

Line 149: Change “all genes” to “every gene” and “build” should be “built.”

Line 154: Remove “subsequently.”

Line 155: Italicize Mastomys coucha.

Line 156: “vial” should be “via.”

Line 157: Remove “Some days” and start sentence with “After.”

Line 160-161: Change to: “After time, larger attached dots without motility…”

Line 161-162: How do you know these cells were in a biofilm? Were any tests performed to demonstrate this?

Line 163: Add “the” before supernatant.

Line 177: Should “PCRs” be primers?

Line 186-187: The “not shown” data should be presented as a table with relevant information that was used to identify R. insidiosa.

Line 192: Change to “a few dead hybridoma…” and should vital be viable?

Figure 2: Is “pure culture” a known control strain, or the black dots purified from the contaminated cell lines? Please clarify this.

Figure 2D: This image is not good quality, and cannot be read. Either provide a higher resolution image, or a summary table with + and – marked.

Line 223-225: Please show the data. It can also be added to a supplemental file. Indicate the number of replicates as well.

Line 223: Are those beta-lactamase genes known carbapenemases? Were any aminoglycoside modifying enzymes identified?

Line 228: Change to “carbapenems.”

Line 232: Why are the breakpoints for Pseudomonas spp., as opposed to another Gram-negative bacterium, used for Ralstonia spp.? A comment in the figure legend or methods should be included.

Discussion: Italicize in vitro and all species names where mentioned.

Line 263: Remove “spp.”

Line 264: Change “Ralstonia spp.” to “species.”

Line 273: Change to carbapenems.

Line 278: Change to “…our R. insidiosa isolate…” and add a comma before “as well as.”

Lines 287-288: Please explain and expand upon what is meant by “single nucleotide polymorphisms <20 nt over the core genome.”

Line 291: Change to “multidrug resistance in…”

Reviewer 2 Report

In the reviewed article, Authors described contamination of hybridoma cell culture by Ralstonia insidiosa. Based on ATCC data, it is estimated that 5–30% of all cell cultures are contaminated with Mycoplasma species. Second common is viral contamination of cell lines. However, important are also contaminations caused by other bacteria, then Mycoplasma, and by fungi. Authors observed  “black dots” in hybridoma cell culture and used Gram staining, 16S rRNA gene sequencing, species-specific PCRs, whole genome sequencing (WGS) and MALDI-TOF MS for identification of contaminant. Moreover, identified Ralstonia insidiosa was eradicated with antibiotics. Authors presented very important problem, frequently occurring in laboratories and I think article will be very interesting for readers. Methodology is correctly described and the results are well presented.

Author Response

We thank the reviewer for his positive feedback on our manuscript.